# Mobile applications available in Saudi Arabia for the management of Primary Dysmenorrhea: A quality review and content analysis

**Reem M. Alwhaibi**[1], **Halla H. Alrwaily**[1], **Fadiah A. AlJaloud**[1], **Sarah A. AlOfaisan**[1], **Norah F. AlSutami**[1], **Haneen M. AlEssa**[1], **Faisal M. Alessa**[2], **Ruqaiyah Khan**[3], **Tahani J. Alahmadi**[4]*

**1** Department of Rehabilitation Sciences, College of Health and Rehabilitation Sciences, Princess Nourah bint Abdulrahman University, Riyadh, Saudi Arabia, **2** Department of Mathematics, King Fahd University of Petroleum, Dhahran, Saudi Arabia, **3** Department of Basic Sciences, Deanship of Preparatory Year, Princess Nourah bint Abdulrahman University, Riyadh, Saudi Arabia, **4** Department of Information Systems, College of Computer and Information Sciences, Princess Nourah bint Abdulrahman University, Riyadh, Saudi Arabia

* Tjalahmadi@pnu.edu.sa

## Abstract

### Background

Primary dysmenorrhea (PD), common in women below 25 years, occurs as pain in the absence of any identifiable pelvic pathology. Menstrual tracking applications (MTAs) may help women manage their PD symptoms. No systematic assessment has been performed on MTA quality with respect to physical therapy management exercise.

### Objectives

This study evaluated the quality of MTAs available in Saudi Arabia for mobile users in both the App Store and Google Play Store and assessed the quality and completeness of exercise regimens provided in these apps using the FITT principle as a guideline for managing PD symptoms.

### Methods

In this cross-sectional study, apps were collected from the App Store and Google Play Store using two strategies for each store independently: Scraper and SimilarWeb. The app quality was evaluated using the Mobile Application Rating Scale (MARS), and exercise content was evaluated based on the recommended Frequency, Intensity, Time, and Type (FITT) principles.

**Data availability statement:** All relevant data are within the paper and its Supporting Information files. We have provided all the data in the manuscript and the supplement files, which are sufficient to replicate the study. The supplements have the data based on which the calculation was done for comparing different applications.

**Funding:** This research was funded by Princess Nourah bint Abdulrahman University, PNURSP2025R117.

**Competing interests:** The authors have declared that no competing interests exist.

## Results

Final evaluation included 16 apps, of which 87.5% required subscription. The mean app quality score ranged from 2.54 (worst-rated app) to 4.45 (best-rated app) with a mean score of $3.54 \pm 0.58$. In addition, only three apps provided all the FITT components in the exercise content.

## Conclusion

This study assessed the quality of exercise provided within these applications as interventions for managing PD symptoms. This evaluation contributes to the understanding of mobile health technologies for PD management in the region, and highlights areas for improvement in app development and content quality to better serve individuals with PD.

## Introduction

The use of mobile apps that can help monitor, advise, or provide interventions for health conditions is known as mHealth [1]. As mHealth technology evolves, it is expected to empower patients to play a more active role in managing their own health. They provide an overarching infrastructure to support self-management, health monitoring, and self-directed learning [1,2]. Furthermore, menstrual tracking applications (MTAs) are widely used in women's health, and their market is estimated to be over $2.5 billion in 2031 [2,3]. It is anticipated that the menstrual health app market will continue to grow from 2023 to 2031, accounting for 40% of total revenue [2]. MTAs are used for various reasons including remembering, predicting the period, tracking symptoms, and seeking more knowledge to understand specific symptoms of dysmenorrhea [3,4]. Likewise, MTAs can be used to identify bleeding issues, prepare for upcoming periods, learn more about menstrual cycles, and facilitate conversations with healthcare providers [5,6]. Moreover, some women use MTAs to record and manage menstrual-related symptoms, including symptoms of premenstrual syndrome and dysmenorrhea.

### Dysmenorrhea and its management

Dysmenorrhea is defined as painful cramps that occur during menstruation and is classified as primary or secondary dysmenorrhea [6]. Primary dysmenorrhea (PD) is common in women below the age of 25 years and occurs as pain in the absence of any identifiable pathology of the pelvis. In a recent systematic review and meta-analysis, the prevalence of dysmenorrhea was 78.5% among young women [7]. The symptoms of PD are cramps and suprapubic pain characterized by colicky spasms, ranging from 8 to 72 h of menstruation and peaking as menstrual flow increases within the first few days [6]. There are different treatment approaches for managing PD that are directly aimed at relieving pain. These include the pharmaceutical approach of prescribing non-steroidal anti-inflammatory drugs(NSAIDs) (e.g.,

ibuprofen) and hormonal drugs [8]. Nevertheless, some side effects of the pharmaceutical approach have been reported, such as prolonged use of NSAIDs associated with stomach ulcers and the use of contraceptives associated with the frequency of bleeding and weight gain [8].

The decision to focus exclusively on PD was based on several reasons. First, PD is far more prevalent than secondary dysmenorrhea (SD) among women under 25, with global prevalence rates ranging from 65% to over 89% [9–12]. Second, PD is characterized by menstrual pain in the absence of pelvic pathology, which makes it amenable to self-management strategies such as exercise [13]. In contrast, SD often requires clinical diagnosis and medical intervention, making it less suitable for mHealth self-care models. This methodological clarity allows for consistency in app content evaluation across a defined population.

**Physical therapy approach for PD management.** Other approaches, such as the physiotherapeutic approach, including exercise-based intervention, have been reported to be effective in reducing PD pain and have the advantage of being harmless, affordable, and cost-effective [14]. A systematic review and meta-analysis showed that exercise, whether low-intensity like yoga or high-intensity like aerobics, can significantly alleviate menstrual pain intensity by approximately 25 mm on a 100 mm VAS [6]. Exercise is recommended for 45–60 min per day three times a week or more, regardless of intensity. Several physical therapy (PT) methods have been found to be effective in improving symptoms, including massaging the area above the pubis with lavender oil, isometric exercises of the adductor and lower abdominal muscles, aerobic exercises such as dancing, yoga, progressive relaxation exercises, stretching, Kegel exercises, jogging, and relaxation exercises [8].

The emphasis on exercise-based interventions in this study stems from several studies highlighting their role in reducing menstrual pain intensity. Exercise, as a physical therapy modality, is non-invasive, cost-effective, and has minimal adverse effects compared to pharmacological interventions. Systematic reviews and meta-analyses [15–18] confirm that both aerobic and stretching-based exercises—such as yoga, walking, and Kegel exercises—reduce pain by approximately 25 mm on the Visual Analogue Scale (VAS). This makes exercise a practical and scalable intervention to be integrated into menstrual tracking apps, which are increasingly used for self-management of dysmenorrhea.

**MTA-a helpful tool for women: Evidence from previous studies.** Menstrual cycle applications can serve as a useful tool to help women reach these criteria for managing menstrual-related symptoms. Women's health-related apps have been evaluated by various researchers over the last decade. Some studies have assessed MTA in different aspects of gynecology, including the ability to support women trying to conceive and predict ovulation [19,20]. However, other studies have evaluated the accuracy of predicting menstrual cycle dates and recording menstrual symptoms [21–23]. Recently, Kalampalikis et al. conducted a systematic review of 14 papers evaluating a single MTA to assess the ability of the app to promote gynecological health. All studies included in the systematic review involved users who downloaded a single selected MTA based on the study's aim, and then the data were collected and analyzed. Kalampalikis et al. concluded that MTAs could play a valuable role in promoting women's health; however, their content requires thorough evaluation by professional healthcare professionals [4]. A recent study by Karasneh et al. systematically reviewed MTAs available on Google Play Store and Apple Store for their quality, using the Mobile Application Rating Scale (MARS) (data in S1 Text). The study concluded that MTAs offer cost-effective solutions for symptom monitoring and management. However, this study did not evaluate how MTAs manage menstruation-related symptoms [24].

Furthermore, a study by Moglia et al. aimed to systematically evaluate MTAs in the Apple store in terms of accuracy, functionality, and features to guide healthcare providers in evaluating and recommending apps for patients. However, this study did not comprehensively address menstrual-related symptom tracking and management features comprehensively [22]. Based on previous literature, the Apple store categorizes the reported symptoms provided by MTAs into body metrics, period-related metrics, cervical states, physical symptoms, and emotional health. Physical symptoms are the largest category of symptoms provided by MTAs(23).

Moreover, over-reporting of these symptoms can lead users to experience a phenomenon called "tracker-fatigue" [4]. MTAs can empower women by teaching them about their bodies and expanding their understanding of menstruation beyond cultural norms (stigma and shame connected with menstruation) [25]. To encourage users to use management methods for controlling menstruation symptoms, educational material features should be made available in MTAs.

**Recent advancements in MTA: Shading light on AI apps and devices.** Recent technological advancements have significantly enhanced the capabilities and user experience of MTAs [26]. Machine learning algorithms are currently employed to improve the accuracy of period predictions and symptom analysis [27]. For instance, Flo Health Inc. has implemented artificial intelligence-driven cycle predictions that adapt to individual user patterns over time, resulting in more personalized and accurate forecasts [28,29].

MTAs are increasingly using wearable technologies, for instance, apps like Ava and Bellabeat track skin temperature, resting pulse rate, and sleep habits. Comprehensive menstrual cycle and fertility insights are available from these data points and user inputs [30,31]. Telemedicine in MTAs is another major development, applications like Clue, Flo and Glow allow users to share menstruation data and get virtual consultations with healthcare specialists [32]. In terms of industry trends, there is a growing focus on inclusivity and diversity in MTA design and functionality. Apps such as Clue and Flo have expanded their features to cater to a wider range of users, including those with irregular cycles, PCOS, endometriosis, and individuals undergoing sex transitions [33,34]. MTAs also add mental health help, and now measure mood, provide mindfulness exercises, and help manage monthly mood fluctuations, thus offering a complete solution.

Hence, the latest trends reveal a shift toward evidence-based app development, as MTAs are now collaborating with doctors and academics to ensure that their information is scientifically correct and therapeutically useful. This development matches the increased desire for digital health solutions to supplement traditional healthcare [35,36]. As menstrual health technology evolves, MTAs can provide complex, tailored, and comprehensive support for menstrual health management.

**Rationale of this study.** With its technologically advanced population, Saudi Arabia has recently experienced a surge in mobile health applications [37,38]. With the rise of digital health solutions, MTAs are expected to gain in popularity among Saudi women. The existing cultural sensitivities and limited public discourse on menstrual health may limit women's access to accurate information and effective symptom management strategies, making systematic evaluations of MTAs in the context of evidence-based physical therapy and Frequency, Intensity, Time and Type (FITT) principles. Effectively designed menstrual tracking applications could assist Saudi women in discreetly and conveniently managing their menstrual health by considering cultural sensitivities. This study aligns with Saudi Arabia's Vision 2030, which advocates women's health and empowerment. This research employed the Mobile App Rating Scale (MARS) to assess Saudi Arabian MTA's and exercise interventions for primary dysmenorrhea symptoms, aiming to address knowledge deficiencies and guide app development, healthcare policies, and women's health initiatives (data in S1 Text). This study focused on the evaluation of the quality of MTAs available to users in Saudi Arabia on both iOS and Android platforms and evaluated the overall quality of these applications and analyzed the extent to which the exercise-based content aligns with evidence-based guidelines, particularly the FITT (Frequency, Intensity, Time, and Type) principles, for managing PD symptoms. Therefore, this study aimed to conduct a dual-layered evaluation: (1) assess the quality of menstrual tracking apps (MTAs) available in Saudi Arabia using the validated Mobile Application Rating Scale (MARS); and (2) evaluate the exercise-based content in these apps to determine its alignment with evidence-based physical therapy guidelines for managing primary dysmenorrhea, with specific reference to the FITT principle.

## Materials and methods

### Study design and setting

A cross-sectional study was performed to assess the quality of mHealth apps for menstrual cycle tracking that are available on two major mobile operating systems: Google Play Store (Android) and Apple Store (iOS). The study methodology was based on five recommended steps for the quality assessment of mHealth apps [39]. Step 1: Select the appropriate

category and number of assessors to assess the quality of the apps; Step 2: Determine the appropriate usage duration for the apps before performing quality assessment; 3: download the app onto the device; 4: Choose an appropriate quality assessment tool; and 5: Assess the app and interpret the results.

## Search strategies for data collection

Two strategies were used to search for MTAs in each store. First, the app-store-scraper [40] and google-play-scraper [41] were used to scrape each store independently for a comprehensive search of MTAs. The second step was to use a web analytical tool called (SimilarWeb) for a specific search of the best-ranked MTAs in the KSA in each store independently.

**Store-scraper.** Raw data from the scraper were collected based on keywords provided by clinical experts: language, support English, Arabic, or both. Keywords are App ID, Title (using "period tracker", "menstrual tracker", "menstrual calendar", "menstrual cycle tracker", "menstrual cycle calendar"), score, developer, last update date, description, number of raters, languages, URL, and price. Permission to collect these data was sought by contacting the support team from both Google and Apple. Because searches using different keywords can lead to duplicated apps, filters were applied to obtain unique apps. These apps had metadata associated with them, as provided by the publisher.

The selected-features that allowed us to perform this study are presented in (Table 1). For comparison with the Google-Play dataset, we selected the number of ratings to show the popularity of the app in both datasets instead of the number of downloads, because this information is not present in the app store. Google Play has a new feature where they only provide the most related apps to any keyword search; this has been confirmed with the Google team. This resulted in fewer apps compared with the AppStore dataset.

**Ranked apps.** The second strategy was to rank the apps using a Similar Web, which is a web analytical tool that has access to various data sources, including internet content and millions of websites and apps. Data collection on a similar web employs an automated technique for capturing and indexing public data from 100 million website pages and 4.7 million applications every month [42]. For searching "Health & Fitness" was selected as the category of apps, and the search term was only "Period" since the web allows using only one word. We chose period as the key term because it is the most popular word. The purpose of this study was to identify the top-ranking MTAs used in *Saudi Arabia* to cover and collect more apps.

## Eligibility criteria

To ensure alignment with primary dysmenorrhea, apps were selected based on metadata and descriptions that referred explicitly to menstrual pain or cramps without referencing diagnosed pelvic pathologies (e.g., endometriosis, fibroids). Apps focused on fertility, ovulation, or general women's health without reference to PD symptom management were excluded. This ensured homogeneity in the condition targeted by the app content.

A complete list of apps from both mobile operating systems was initially screened, based on their titles and descriptions. Apps meeting the following criteria were selected: apps related to dysmenorrhea and included exercise as an intervention, supporting either Arabic or English languages or both, and apps that did not require any external device. Although the search aimed to include apps with Arabic language support, the majority of high-rated apps available in Saudi Arabia only offered content in English. Arabic availability was not used as an exclusion criterion to avoid omitting high-quality tools used by bilingual users. An average user rating of ≥ 4 out of 5 indicated that apps with high standard

**Table 1. Selected features\characteristics of apps in data collection.**

| Feature | App ID | Title | Score | Developer | Last update date | Description | The number of raters | Languages | URL | Price |
|---|---|---|---|---|---|---|---|---|---|---|
| App-Store | P | P | P | P | P | P | P | P | P | P |
| Google-Play | P | P | P | P | P | P | P | O | P | P |

user ratings were more frequently downloaded. To ensure accuracy, free and paid apps with or without subscription were selected if they were published or updated since 2018. To ensure alignment with PD, apps were included only if their metadata and descriptions explicitly referred to menstrual pain or cramps, without referencing diagnosed pelvic pathologies such as endometriosis or fibroids. Apps that focused solely on fertility tracking, ovulation, or general women's health without reference to menstrual pain management were also excluded. This ensured the study maintained focus on PD as a self-manageable condition and avoided content related to secondary dysmenorrhea. The apps were excluded if they were duplicated, had technical problems, or had no exercise content for pain management in PD patients. Other exclusion criteria were based on being outdated (non-functional), apps with ratings below four, apps not rated at all, or language limitations. A total of 598 apps were excluded, the details of which are illustrated in the PRISMA flowchart (Fig 1).

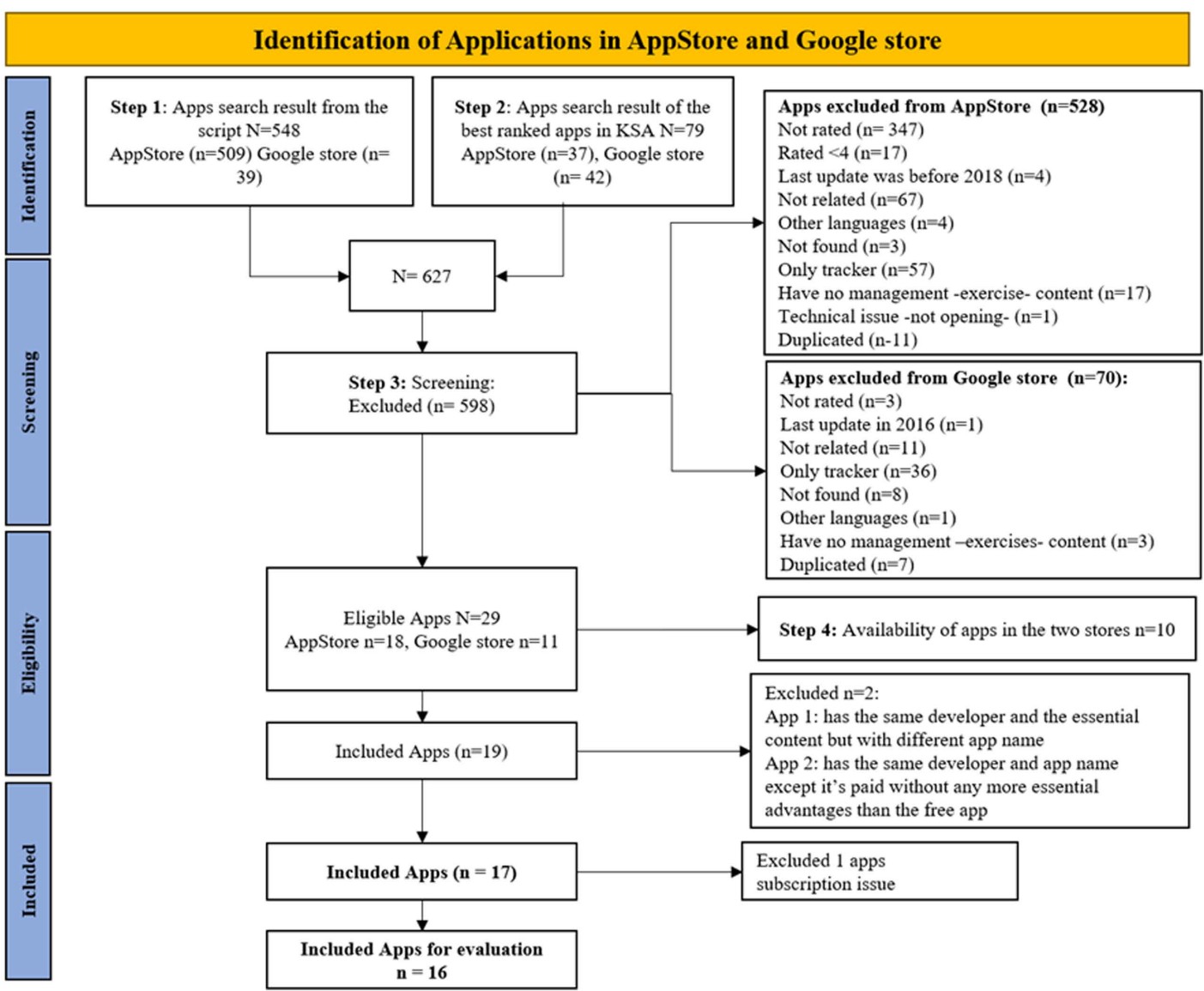

**Fig 1. PRISMA flowchart of search results.**

## App quality assessments

Two evaluators (SA and NF) independently evaluated each app using the Mobile Application Rating Scale (MARS) tool (data in S1 Text) [43]. Any disagreement was resolved by discussion with a third evaluator (HH) to reach a consensus on each rating for every MARS subscale. Before evaluating the apps, all evaluators watched the MARS training video using the original developer of the MARS tool [43]. In addition, a training session on the MARS evaluation process was conducted on an app excluded from the review, and the results were discussed to ensure that all evaluators understood the MARS items and rating process.

The MARS tool is designed to assess the quality of mHealth applications using multidimensional measures, including qualitative, objective, and subjective aspects. MARS tool was chosen because it has good to excellent reliability (internal consistency Omega = 0.79 to 0.93) and an inter-rater reliability of ICC = 0.82 [44]. It has also been validated and widely used to rate the quality of mHealth apps [45–49].

MARS is structured into three categories: app classification to collect descriptive information about the apps. The second category is the app quality rating category involving 23-items subdivided into two dimensions: objective and subjective quality. The objective quality dimension consists of 19-items categorized into four subscales (engagement, 5 questions; functionality, 4 questions; aesthetics, 3 questions; and Information Quality, 7 questions). The subjective quality dimension consists of four questions. Each item is rated on a 5-point Likert scale. The third category is app-specific, which is used to assess the impact of apps on users' behavioral change and consists of six items rated on a 5-point Likert scale. An online survey by SurveyMonkey was used to help the evaluators complete the MARS.

## Data extraction and app content analysis

The data extraction process used in this study involved two evaluators who assessed the 16 MTA apps. The evaluation is conducted in two sections. The first section concentrated on general app information, including responses to inquiries such as: Does the app require the user to create an account before granting access to the app content; does the app ask for personal information such as name, age, date of birth, height, weight, and menstrual cycle; does the app have terms and conditions or a disclaimer; and does the app provide a list of developer qualifications?

The second section was devoted to evaluating the recommended exercise content using the FITT concept, namely, how the activities were presented (via audio, spoken word, written text, visual images, or video). This idea underpins the precise delivery of the workout recommendations. Exercise programs should incorporate the major training components of the FITT principle (F:frequency, I:intensity, T:time, and T:type) [50,51], benefits, recommendations, and contraindications to exercise according to the ACSM Guidelines of Exercise Testing and Prescription. Any disagreement between the two evaluators was resolved by discussion and then by a third evaluator.

## Data analysis

Descriptive statistics were used to analyze the data obtained from the data extraction and evaluation of the apps using the MARS tool. The mean score for each subscale of the objective quality dimension in the second category (engagement, functionality, aesthetics, and information quality) and the overall mean score were calculated. In addition, the mean scores of the subjective quality dimension items of the second category and the app-specific items of the third category were calculated. Additionally, the app overall mean score was categorized into tertiles (best T1, average T2, or worst quality app T3). Based on the tertile range of the MARS score which is 5\5 the range is as following (T1 = 3.35–5, T2 = 1.68–3.34, or T3 = 0–1.67). The analysis was performed using Microsoft Excel 2022 software.

# Results

## App selection process

Searches on the App Store and Google Play Store using scripts and the best-ranked KSA apps yielded 627 apps for screening. The initial screening excluded 598 (95.37%) apps that were either not rated or were rated under four, unrelated to the period trackers, were in languages other than English or Arabic, were last updated before 2018, were not found in the stores, were only trackers, or the content did not include exercise management, had technical issues, or were duplicated apps. Of the remaining 29 (4.63%) apps, 10 (1.59%) were duplicated apps, and 2 (0.32%) were excluded as one of them had the same developer and the essential content but with different app names, and the other app had the same developer and app name except for a paid app without any more essential advantages than the free app. Finally, one (0.16%) app was excluded due to subscription issues during the evaluation period. Therefore, the final number of apps included in the evaluation was 16 (2.55%). Fig 1 shows a flowchart of the search process.

## App characteristics and classification

Of the 16 reviewed apps, eight (50%) could be downloaded from the App Store, one (6.25%) could only be found on Google Play, and seven (43.75%) could be accessed from both the App Store and Google Play. Each software package could be downloaded for free, but 14 (87.5%) required subscriptions. The average star rating of the app when the investigators downloaded it was 4.61. The characteristics of the apps used in this study are listed in Table 2.

The MARS tool was used for app classification. First, 75% of the apps had commercial objectives and were affiliated with non-governmental groups. None of them worked governmental minutes in academic institutions. All apps had a physical health focus, with 93.8% promoting happiness and well-being and 93.8% encouraging mindfulness, meditation, and relaxation. Monitoring and tracking were the theoretical underpinnings or tactics used across all apps, followed by

**Table 2. Characteristics of the apps included (N = 16).**

| Store and app ID | App name | Developer | Basic | Upgrade | Update | Version | Rating | Ratings, n |
|---|---|---|---|---|---|---|---|---|
| App Store | | | | | | | | |
| 01 | Period Tracker Period Calendar | Abishkking limited | Free | Paid | 2022 | 2.50.1 | 4.9 | 15348 |
| 02 | Period Tracker, Cycle Tracking | Real Vision LTD. | Free | Free | 2022 | 1.1 | 5 | 1 |
| 03 | Cycle and Period Tracker – Femi | Magicfit Limited | Free | Paid | 2023 | 1.16.0 | 5 | 2 |
| 04 | Filo Ovulation Period & Tracker | Kipkemoi Ibrahim | Free | Paid | 2023 | 3.4 | 5 | 1 |
| 05 | Teen Period Tracker | Vipos.com | Free | Paid | 2022 | 1.3.2 | 5 | 3 |
| 06 | Period Tracker | Luni | Free | Paid | 2022 | 2.0.6 | 4 | 4 |
| 07 | Paloma Period Tracker & Diary | Apalon Apps | Free | Paid | 2020 | 1.1 | 4.4 | 66 |
| 08 | Period Tracker Plus | Flatcracker Software | Free | Paid | 2022 | 10.96 | 4.5 | 134 |
| Google Play Store | | | | | | | | |
| 09 | Ovulation and Period Tracker | Leap Fitness Group | Free | Paid | 2023 | 1.089.GP | 4.9 | 383667 |
| App Store and Google Play Store | | | | | | | | |
| 10 | Clue Period & Cycle Tracker | BioWink | Free | Paid | 2023 | 109.0 | 4.7 | 3231 |
| 11 | Femometer Fertility Tracker | Bongmi Global Group | Free | Paid | 2023 | 5.25.5 | 4.8 | 87 |
| 12 | Flo Ovulation & Period Tracker | Flo Health Inc. | Free | Paid | 2023 | 9.21.0 | 4.7 | 13954 |
| 13 | Glow: Fertility and Ovulation App | Glow | Free | Paid | 2023 | 9.8.9 | 4.8 | 148 |
| 14 | WomanLog calendar | Pro Active App | Free | Paid | 2023 | 6.8.2 | 4.8 | 23 |
| 15 | Ovia: Fertility, Cycle, Health | Ovia Health | Free | Free | 2023 | 5.12.0 | 4.6 | 133 |
| 16 | Period Diary Ovulation Tracker | Bellabeat, Inc. | Free | Paid | 2023 | 5.0.8 | 4.7 | 2133 |

information and education. Finally, the most frequent technical aspect of the apps was sending reminders (93.8%), and 68.8% of the apps allowed password protection (data in S1 Text).

## Quality assessment

The specific scores for each application are presented in Table 3 and Fig 2. The mean app quality score ranged from 2.54 (worst-rated app) to 4.45 (best-rated app), and a similar situation was observed in each section: 1.4 to 4 (engagement), 3.25 to 5 (functionality), 2–5 (aesthetics), 2 to 4.4 (information), 1.63 to 3.22 (app subjective quality), and 1.17 to 4.83 (app specific). Overall, the 16 apps obtained a mean score for the quality of (3.54±0.58). On average, the best-rated section was functionality (4.30±0.50), followed by aesthetics (3.75±0.95) and information (3.32±0.71), whereas the worst-rated sections were engagement (2.8±0.85), app subjective quality (2.83±0.42), and app-specific section (2.72±1.17).

## Data extraction

**Personal Information, Information regarding period, Terms & Conditions/Disclaimer, Developer qualifications, Reference list & Sources of information.** Only 6 (37.5%) of the 16 apps demanded that users sign up for accounts. Eleven (68.75%) applications requested the user for personal information (such as age, height, and weight), whereas five (31.25%) websites did not. In terms of period-related information, 13 (81.25%) of the 16 apps requested data on average cycle length, average menstrual cycle length, cycle regularity, menstrual symptoms, and the last period date. Cycle regularity was requested by 8 (50%) patients, menstrual symptoms by 7 (43.75%), and last period by 12 (75%). Finally, 14 apps (81.25%) included a disclaimer or set of terms and conditions that released the app's creators from liability (i.e., user participation is at their own risk and the app bears no responsibility for any unfavorable outcomes that may occur during use of the app or as a result of using the app). In eight (or 50%) of the apps, users had to willingly accept the terms and

**Table 3. Mobile App Rating Scale (MARS) scoring for MTAs.**

| App ID | App name | Overall app quality Mean ±SD* | T | Section means | | | | | |
|--------|----------|-------------------------------|---|------|------|------|------|------|------|
| | | | | A | B | C | D | E | F |
| 01 | Period Tracker Period Calendar | 4.10±0.70 | T1 | 3.6 | 4.75 | 4.66 | 3.4 | 3.13 | 3.50 |
| 02 | Period Tracker, Cycle Tracking | 2.95±1.16 | T2 | 1.4 | 4 | 3.66 | 2.75 | 1.63 | 1.33 |
| 03 | Cycle and Period Tracker – Femi | 3.94±1.16 | T1 | 2.2 | 4.5 | 4.66 | 4.4 | 2.41 | 4.17 |
| 04 | Filo Ovulation Period & Tracker | 2.75±1.15 | T2 | 1.4 | 4 | 3.33 | 2.25 | 1.90 | 1.33 |
| 05 | Teen Period Tracker | 2.54±0.88 | T2 | 1.8 | 3.75 | 2 | 2.6 | 1.86 | 1.17 |
| 06 | Period Tracker | 3.85±0.87 | T1 | 2.8 | 4.75 | 4.33 | 3.5 | 2.57 | 3.50 |
| 07 | Paloma Period Tracker & Diary | 3.44±0.83 | T1 | 2.6 | 4.5 | 3.66 | 3 | 2.04 | 2.17 |
| 08 | Period Tracker Plus | 3.06±1.23 | T2 | 4 | 4.25 | 2 | 2 | 2.58 | 3.00 |
| 09 | Ovulation and Period Tracker | 3.95±0.43 | T1 | 3.4 | 4.25 | 4.33 | 3.8 | 2.25 | 3.17 |
| 10 | Clue Period & Cycle Tracker | 3.63±1.16 | T1 | 2 | 4.75 | 4 | 3.75 | 2.48 | 2.17 |
| 11 | Femometer Fertility Tracker | 4.20±0.82 | T1 | 3.8 | 4.75 | 5 | 3.25 | 2.72 | 2.67 |
| 12 | Flo Ovulation & Period Tracker | 4.14±0.48 | T1 | 3.6 | 4.75 | 4 | 4.2 | 2.23 | 4.83 |
| 13 | Glow: Fertility and Ovulation App | 2.97±0.30 | T2 | 3.2 | 3.25 | 2.66 | 2.75 | 2.45 | 1.33 |
| 14 | WomanLog calendar | 3.30±0.49 | T2 | 2.6 | 3.75 | 3.33 | 3.5 | 2.32 | 2.67 |
| 15 | Ovia: Fertility, Cycle, Health | 3.41±0.45 | T1 | 2.8 | 3.75 | 3.33 | 3.75 | 2.40 | 2.00 |
| 16 | Period Diary Ovulation Tracker | 4.45±0.68 | T1 | 3.6 | 5 | 5 | 4.2 | 3.22 | 4.50 |

A, engagement section; B, functionality section; C, aesthetics section; D, information section; E, subjective quality; F, app-specific section; T1, best app quality; T2, average app quality; T3, worst app quality.

*Overall mean included only A, B, C, D sections from MARS.

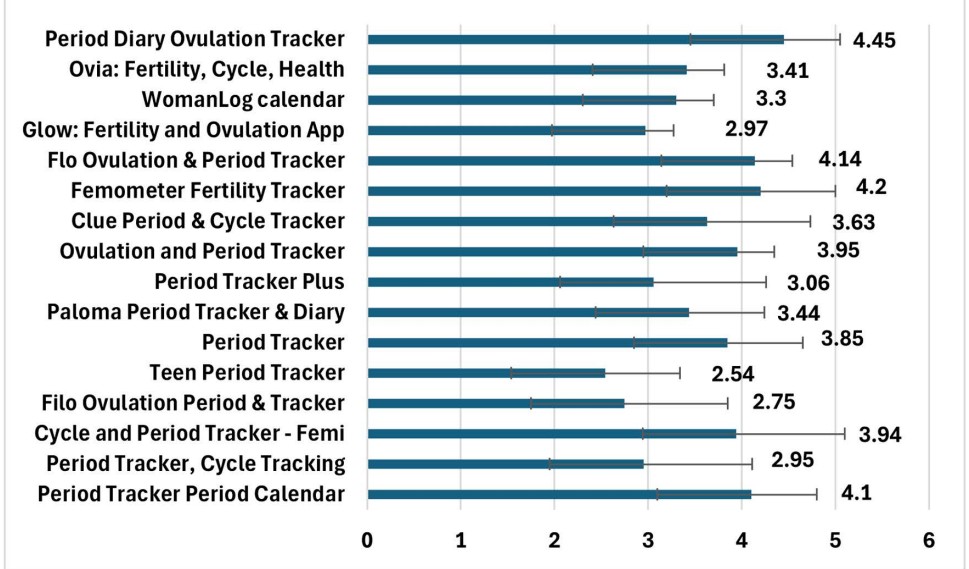

**Fig 2. Shows the comparison of mean app quality score.**

conditions. Only 2 (12.5%) offered a list of developer requirements. Nine apps (56.25%) offered a reference list or based their information on sources; academic literature was the source that was most frequently used (data in S2 Text).

### Exercise and screening for contraindications

Only 3 (18.75%) included all four forms of cues for the elements used to display or explain the exercises: spoken, textual, still images, and videos. One software program included three cues: written, video, and still images. Three apps offered both written and still image clues, whereas only one app offered video prompts. Of the apps, 6.25% offered verbal and written cues and seven apps exclusively offered written prompts.

Only 3 (18.75%) apps delivered all FITT components according to the FITT principle. Except for the intensity, 2 (12.50%) other apps offered all FITT components. Time and type were provided by two (12.50%), and type and intensity were provided by one (6.25%) app. While 6 (37.5%) apps simply offered typing. Only intensity was provided by 1 (6.25%) app. One (6.25%) of the apps did not offer any FITT primary components.

Only 1 (6.25%) of the apps offered the necessary degree of experience. For workouts, two (12.50%) apps required equipment. Eleven (68.75%) apps offered yoga, stretching (56.25%), Pilates (18.75%), flexibility (6.25%), HIIT (6.25%), dancing (0.00%), and other forms of exercise (0.00%). Whereas, 10 apps offered walking, jogging, or running; meditation (37.50%); breathing exercises (31.25%); pelvic floor/Kegel exercises (12.50%); cycling (18.75%); swimming (12.50%); resistance training (6.25%); and weight training (0.00%). During this time, 16 (100%) apps improved physical symptoms, and 15 (93.75%) apps improved mental symptoms as a result of the health advantages of exercise. None of the apps requested exercise for any contraindications. Of these applications, 62.50% did not offer workout advice (data in S3 Text).

### Discussion

This study contributes novel insights into the landscape of menstrual tracking applications available in Saudi Arabia, particularly regarding their adherence to evidence-based physical therapy strategies for managing PD. Unlike prior app reviews that focused solely on user ratings or app functionality, this study integrates a clinical perspective by examining whether apps offer structured, guideline-based exercise protocols using the FITT framework. The results reveal that while

technical and aesthetic quality was generally acceptable, there is a striking deficiency in medically grounded, structured exercise content designed for dysmenorrhea relief.

The present descriptive study was the first to investigate the quality of MTAs' workout content and provide thorough feedback on the app's information from a physical therapist's viewpoint. Statistics suggest that 980 and 667 new programs were added to the Google Play Store and App Store, respectively [52]. The present study screened 16 apps from these stores, but not all the reviewed apps were offered for free; the majority had in-app purchases and required an Internet connection to function. This was in contrast to a previous study conducted to evaluate MTAs quality and effectiveness, which included 49 MTAs and focused only on free-period tracker apps [24]. However, the current study focused on searching for all high-quality apps despite their cost, which varied between SAR 5.99–62.99 for monthly subscriptions. This indicated a lack of organizational monitoring for app costs, as this range was inconsistent with app quality; some low-quality apps have high cost and vice versa. For example, Filo Ovulation & Period Tracker had low quality compared to other reviewed apps, and it had a monthly subscription of SAR 42.99. Previous studies that assessed the quality of MTAs did not include apps with paid subscriptions [22,24,53], which limited the scope of the current study's additional examination of MTAs' costs.

Another important aspect is the language supported by the apps. Of the apps reviewed, only two supported both languages (English and Arabic). However, none of them displayed any content when the language was changed to Arabic. This may be a significant barrier in helping Arabic women learn more about their menstrual cycles or even discuss exercises to assist in reducing their PD symptoms.

Additionally, more than one-third of the apps had no reference list for their information sources. This issue was also reported in a study that evaluated physical activity apps for pregnant women, where they found only one app (out of 27) that provided a reference list to support the content of the app [54]. Additionally, a systematic review of 52 studies that covered 6520 applications revealed that only 10–87% of 814 apps effectively addressed medical information. This study examined mHealth apps for expert involvement and adherence to evidence-based sources [55]. Trusting such content and information may be detrimental to people's health, as experts are not involved in the app's creation or evaluation process.

The average app quality score for the sixteen total apps was 3.54/5, which was better than the average subjective part score of 2.387/5 for the MARS tool's overall mean scores. Two factors may account for this discrepancy. First, the Engagement part of the app quality score had the lowest mean score of the four, which was thought to have resulted in the users' lack of interest and interaction. This explains the low subjective quality mean score of the app. This justifies the low results of the subjective quality mean score for the app. Additionally, most applications did not offer material in Arabic or consider the cultural or religious preferences of the target audience, which resulted in the engagement section's mean score being low. Second, the Functionality area had the highest mean score, which may have caused the mean score for the quality of the app to be greater than the mean score for the subjective quality of the app. The overall high score of the Functionality section could be attributed to the fact that the evaluators lacked sufficient information technology (IT) background to address technical issues, or any specific components related to the apps' functionality and performance as an IT professional would view and address them accurately.

These findings are consistent with a prior study that used the MARS technique to evaluate the quality of 119 MTAs and found that the functioning mean score was the highest, whereas the engagement mean score was the lowest [53]. This demonstrates the necessity for MTAs to improve user engagement.

The scores of the included programs, as determined by the findings of the current study, ranged from 2.54 to 4.45, with Period Diary Ovulation Tracker having the highest score and Teen Period Tracker having the lowest. These findings differ from those of a previous study that assessed MTAs and discovered that the Teen Period Tracker application, along with a few other applications such as the WomanLog calendar, Clue Period & Cycle Tracker, and Glow: Fertility Ovulation App scored higher than in the present study [24]. The distinction in goals between the current study and the prior study is thought to be the cause of these discrepancies. Additionally, because the previous study lacked scores for each

component, we were unable to compare the section scores between the current study and the previous study to determine how the scores varied [24].

The second goal of this study was to examine the exercise content of MTAs in order to determine whether the apps offered high-quality exercise prescriptions in accordance with the FITT guidelines suggested by the ACSM [50,51]. The FITT principle serves as a dose-based instruction manual for performing activities. Most of the apps we tested included FITT information. However, only 18.75% (3 out of 16) of the apps included all the FITT principles to direct users who were performing these activities. In addition, not every premise has been addressed by other applications.

In these apps, the type of exercise was the most common component at 87.5% (14 out of 16 apps). Stretching (56.25%) and yoga (68.57%) were the most advised forms of exercise, followed by cardiovascular activity (walking, jogging, or running) in 9 out of 16 apps. This is in line with recent research showing that stretching, yoga, and aerobics can help reduce the symptoms of PD [6,8,56]. Although Zumba dance has been shown to be useful in reducing menstrual pain in young women, none of the assessed apps suggest dancing as a method of managing periodic pain.

Furthermore, only 5 out of 16 apps offered user information on their activity intensity. However, there is variation in the literature regarding what level of activity is best to lessen the symptoms of PD [6,56]. Consequently, apps that offer workout intensity vary in their recommended intensity throughout their content. This made it difficult to classify apps according to exercise intensity in this study. Despite this, a systematic analysis that examined the effectiveness of exercise in women with PD concluded that any activity, no matter how intense, has a significant impact on reducing the intensity of menstruation discomfort [6]. Despite the fact only 5 of the 16 apps guided the user how intensely they should practice.

However, the frequency, time, and duration of exercise have been documented in literature [6,57,58]. According to the data, exercising for 30–60 min per day, two–three times per week, or more can help reduce PD discomfort [6,57,58]. However, only 31.25% of apps provided users with information on how often to exercise, and only 43.75% of apps provided users with information on how long to exercise. This can be explained by the fact that the reviewed apps were not created by a healthcare professional, vetted by a healthcare professional, or exercise based.

The studied apps' descriptions claimed that they are not only used for tracking the menstrual cycle but also for helping women handle their menstrual concerns, including managing their PD symptoms. This is true even though they are not exercise-based applications. Therefore, apps are not entirely correct in their descriptions when they are unable to provide a thorough explanation of the management approach, including the description of workouts. Considering this, the app's quality may suffer, which can be substantiated by a similar conclusion made in a previous study that showed how few apps were created using an evidence-based approach and how little information was provided in MTAs regarding pain and symptom management techniques [53].

The present study has several advantages, as a thorough search for apps in the AppStore and Google Play stores was carried out using two methods to look for MTAs. In contrast, most research lacks a thorough explanation of the search methodology. Additionally, the study employed the well-accepted, valid, and accurate MARS technique to assess the caliber of the mHealth apps.

## Recommendations

The involvement of health care professionals in the app review process is crucial to ensure that all materials are accurate and suitable for all female groups. To guide the user effectively while exercising, the app's workout content must address each of the FITT major principles. Apps should also support additional languages, such as Arabic, and developers should consider the age group and cultural differences. A scientific evaluation is necessary prior to the app's global release to ensure that it meets the needs of females across different ages and cultures. Health organizations should define and provide explicit criteria for app pricing. The evaluation using the MARS tool would benefit from involving evaluators with diverse backgrounds, including IT specialists in the functionality section. Additionally, the involvement of experts across

relevant fields, such as gynecology, exercise physiology, and physical therapy, could strengthen the study further. The design of an m-health tool specifically for physical therapy evaluation is recommended. Future research should focus on developing and testing MTAs that incorporate evidence-based exercise protocols, particularly for the management of PD. This includes validating the effectiveness of the exercise regimens in clinical trials. This study also emphasizes the importance of involving healthcare professionals, such as physical therapists, in app development. Future research could investigate the impact of interdisciplinary collaboration on app quality and effectiveness.

## Limitations

During the evaluation process, some of the app's ratings in the store changed, and one paid app encountered payment issues related to the app that prevented evaluators from accessing in-app purchases. Hence, the app was excluded from the study because the use of websites was not within the study aim. Although Arabic language support was considered during app selection, most of the included apps were available only in English. This reflects the current market reality, where English-language apps dominate even in Arabic-speaking regions. Additionally, the focus of this study can only be focused to English speaking Saudi population as most of the apps were only in English, limiting the cultural and linguistic relevance of the findings for monolingual Arabic-speaking users. This highlights a broader gap in the availability of culturally tailored, Arabic-language mobile health applications for menstrual health management in the region. This regional focus may limit the generalizability of the findings to other countries or cultures. Only two apps appeared to have been created by healthcare professionals with medical degrees, and neither the apps nor the developers were affiliated with any reputable organizations. Additionally, it was difficult to determine who developed the majority of the apps. On the other hand, the lack of PT guidelines for the treatment of dysmenorrhea limited the study's ability to further evaluate the exercise content by categorizing it according to exercise intensity. Another drawback of this study was the need for the involvement of evaluators with various educational backgrounds, such as those in technology and health, for the MARS to conduct a thorough examination of all factors. Additionally, this study excluded apps rated under 4 stars, which may have eliminated potentially useful apps that did not meet this threshold. This criterion could introduce bias and limit the generalizability of our findings to a broader range of MTAs.

## Conclusion

The study examined period tracking apps available in Saudi Arabia and found that most had significant problems. Although many apps scored well in technical aspects and user interface design, the majority failed to offer structured, evidence-based exercise regimens suitable for primary dysmenorrhea management, and few supported the Arabic language. This study evaluated both the technical quality and clinical relevance of these apps, highlighting a striking deficiency in medically grounded, exercise-focused content. The findings underscore the need for interdisciplinary collaboration between app developers and healthcare professionals—particularly physical therapists—to design tools that are not only user-friendly but also aligned with best practices in women's health. These insights are essential for informing future mHealth innovation, policy recommendations, and patient-centered digital health solutions.

## Supporting information

**S1 Text. App classification using MARS tool.**
(DOCX)

**S2 Text. Personal Information, Terms and Conditions, and Disclaimer.**
(DOCX)

**S3 Text. App exercise content.**
(DOCX)

## Acknowledgments

The authors would like to thank "Princess Nourah bint Abdulrahman University, Researchers Supporting Project number (PNURSP2025R117), Princess Nourah bint Abdulrahman University, Riyadh, Saudi Arabia" for supporting this project.

## Author contributions

**Conceptualization:** Reem M. Alwhaibi, Tahani J. Alahmadi.

**Data curation:** Halla H. Alrwaily, Fadiah A. AlJaloud, Sarah A. AlOfaisan, Norah F. AlSutami, Haneen M. AlEssa.

**Formal analysis:** Halla H. Alrwaily, Fadiah A. AlJaloud, Sarah A. AlOfaisan, Ruqaiyah Khan, Tahani J. Alahmadi.

**Funding acquisition:** Reem M. Alwhaibi.

**Methodology:** Fadiah A. AlJaloud, Sarah A. AlOfaisan, Faisal M. Alessa, Tahani J. Alahmadi.

**Project administration:** Halla H. Alrwaily.

**Resources:** Faisal M. Alessa.

**Software:** Haneen M. AlEssa, Faisal M. Alessa.

**Supervision:** REEM alwhaibi, Reem M. Alwhaibi.

**Validation:** Haneen M. AlEssa, Faisal M. Alessa.

**Writing – original draft:** Halla H. Alrwaily, Fadiah A. AlJaloud, Sarah A. AlOfaisan, Norah F. AlSutami, Haneen M. AlEssa.

**Writing – review & editing:** REEM alwhaibi, Reem M. Alwhaibi, Ruqaiyah Khan, Tahani J. Alahmadi.

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
