## [Decision Letter · Decision Letter 0]

12 Jan 2025

PONE-D-24-49548Mobile applications available in Saudi Arabia for the management of Primary Dysmenorrhea: a quality review and content analysisPLOS ONE

Dear Dr. alwhaibi,

Thank you for submitting your manuscript to PLOS ONE. After careful consideration, we feel that it has merit but does not fully meet PLOS ONE’s publication criteria as it currently stands. Therefore, we invite you to submit a revised version of the manuscript that addresses the points raised during the review process. Please submit your revised manuscript by Feb 26 2025 11:59PM. If you will need more time than this to complete your revisions, please reply to this message or contact the journal office at plosone@plos.org . Please include the following items when submitting your revised manuscript:

We look forward to receiving your revised manuscript.

Kind regards,

Mukhtiar Baig, Ph.D.

Academic Editor

PLOS ONE

“The authors would like to thank “Princess Nourah bint Abdulrahman University, Researchers Supporting Project number (PNURSP2024R117), Princess Nourah bint Abdulrahman University, Riyadh, Saudi Arabia” for supporting this project.”

4. In the online submission form you indicate that your data is not available for proprietary reasons and have provided a contact point for accessing this data. Please note that your current contact point is a co-author on this manuscript. According to our Data Policy, the contact point must not be an author on the manuscript and must be an institutional contact, ideally not an individual. Please revise your data statement to a non-author institutional point of contact, such as a data access or ethics committee, and send this to us via return email. Please also include contact information for the third party organization, and please include the full citation of where the data can be found.

Reviewers' comments:

Reviewer's Responses to Questions

**Comments to the Author**

1. Is the manuscript technically sound, and do the data support the conclusions?

Reviewer #1: Yes

Reviewer #2: Partly

2. Has the statistical analysis been performed appropriately and rigorously? 

Reviewer #1: Yes

Reviewer #2: Yes

3. Have the authors made all data underlying the findings in their manuscript fully available?

Reviewer #1: Yes

Reviewer #2: Yes

4. Is the manuscript presented in an intelligible fashion and written in standard English?

Reviewer #1: Yes

Reviewer #2: Yes

5. Review Comments to the Author

Reviewer #1: You wrote about a relevant paper addressing a possible opportunity to increase women's Health in Saudi Arabia. It was a well-written paper which was pleasant to read.

My main issue is that I am missing a concrete research question and hypothesis. Usually, the introduction ends with a research question, but yours does not. What is your research question, and why is this relevant?

However, I found a few additional issues difficult to distil from the paper, which might increase the paper's quality and better address your point.

1. Why are you focusing on physical exercises after period tracking? You wrote quite a long introduction, but I don't find any solid arguments as to why exercising should be part of the period tracker. Many lifestyle interventions influence dysmenorrhea (e.g. diets, sleep, etc). Why do you want to narrow your evaluation to only exercises?

2. Why are to focusing on primary dysmenorrhea? There are no good arguments why period tracking, combined with exercises, is only effective for primary dysmenorrhea and not, for example, for endometriosis. Please elaborate on why you limit to primary dysmenorrhea.

3. why limit to Arabic? I understand that you want to evaluate the effect on Arabic culture, but most of your apps are in English only. I don’t understand how your methodology addressed the Arabic problems. It would make more sense if you only used Arabic apps, but you didn’t. So why does your study not apply to all English-speaking countries? You did not exclude anything based on the Arabic nature of your study, did you?

4. you introduce the relevance of primary dysmenorrhea, but you don't mention it anymore in the paper in the results or discussion sections. How did you evaluate the apps for primary dysmenorrhea only? How did you know if the app was used for secondary dysmenorrhea either? Please explain the role of primary dysmenorrhea in the evaluation of the apps in the results of method section.

Finally, what is your conclusion? This links back to your question. What is the answer to your question, based on the results? I don’t understand what I should have learned from your paper and how that can influence future research. It seems now that your conclusion is mainly for app developers, but your readers are mainly researchers, not developers.

Reviewer #2: Overall, the manuscript presents a well-executed study with significant contributions to the field of mHealth and women’s health. The methodology is rigorous, and the findings are well supported by data. However, certain areas require further refinement. The research question should be more explicitly stated in the introduction, and justification for methodological choices, such as the exclusion of lower-rated apps, should be provided. Greater emphasis on inter-rater reliability would enhance the validity of the app evaluations. The discussion would benefit from more direct comparisons with prior studies, and the results could be presented more effectively with visual aids. The conclusion should incorporate broader policy implications and clearly define future research directions.

With these refinements, the manuscript would be strengthened in clarity, methodological rigor, and overall impact.

6. PLOS authors have the option to publish the peer review history of their article (what does this mean? ). If published, this will include your full peer review and any attached files.

**Do you want your identity to be public for this peer review?** For information about this choice, including consent withdrawal, please see our Privacy Policy .

Reviewer #1: **Yes: ** R.A. de Leeuw

Reviewer #2: **Yes: ** Dr.Noor-i-Kiran Naeem.

---

## [Author Response · Author response to Decision Letter 1]

6 Mar 2025

Response to Reviewer comments

Authors would like to thank the reviewers for their genuine and sincere comments on our study. We have considered all the comments respectfully and tried our best to address them in the best possible manner. Changes in the manuscript are carried out in review mode to highlight the additions or deletions.

Reviewer #1: You wrote about a relevant paper addressing a possible opportunity to increase women's Health in Saudi Arabia. It was a well-written paper which was pleasant to read.

My main issue is that I am missing a concrete research question and hypothesis. Usually, the introduction ends with a research question, but yours does not. What is your research question, and why is this relevant?

However, I found a few additional issues difficult to distil from the paper, which might increase the paper's quality and better address your point.

1. Why are you focusing on physical exercises after period tracking? You wrote quite a long introduction, but I don't find any solid arguments as to why exercising should be part of the period tracker. Many lifestyle interventions influence dysmenorrhea (e.g. diets, sleep, etc). Why do you want to narrow your evaluation to only exercises?

Response 1: Dear Sir, Thank you so much for your valuable insights. Physical therapy approaches, particularly exercise-based interventions, have demonstrated significant effectiveness in reducing primary dysmenorrhea pain. Both low-intensity exercises like yoga and high-intensity exercises such as aerobics have shown positive results in alleviating menstrual pain intensity. Exercise is highlighted as a safe, affordable, and cost-effective method for managing PD symptoms, making it a practical alternative to pharmaceutical treatments that often come with side effects. Specific exercise regimens, including 45 to 60 minutes of exercise three times a week with activities like yoga, aerobic exercises, stretching, Kegel exercises, and relaxation techniques, have been found to effectively improve PD symptoms. While other lifestyle interventions are acknowledged, exercise-based interventions are emphasized as particularly advantageous due to their safety, affordability, and effectiveness in managing PD symptoms. This is we focused on the evaluation of exercise programs in the apps.

2. Why are to focusing on primary dysmenorrhea? There are no good arguments why period tracking, combined with exercises, is only effective for primary dysmenorrhea and not, for example, for endometriosis. Please elaborate on why you limit to primary dysmenorrhea.

Response 2: Our research limits the evaluation to primary dysmenorrhea because it is defined by the absence of pelvic abnormalities, predominantly affects young women, and allows for a more focused and consistent methodological approach. Although other lifestyle interventions may impact various conditions, the effectiveness of period tracking combined with exercise is evaluated specifically in the context of primary dysmenorrhea due to its high prevalence and the clarity of its diagnostic criteria.

3. Why limit to Arabic? I understand that you want to evaluate the effect on Arabic culture, but most of your apps are in English only. I don’t understand how your methodology addressed the Arabic problems. It would make more sense if you only used Arabic apps, but you didn’t. So why does your study not apply to all English-speaking countries? You did not exclude anything based on the Arabic nature of your study, did you?

Response 3: We appreciate the valid critique because there appears to be a disconnect between the stated focus on Arabic culture/users and the actual methodology and app selection. This is one of the limitations of our research. The study doesn't present clear evidence of why its findings would be specific to Arabic contexts when the apps evaluated were predominantly in English. The lack of Arabic language support in the apps suggests the findings might be more applicable to English-speaking populations. This suggests that the study's results could be more broadly applicable to English-speaking population of the country. Dear Reviewers, as you know the study was carried out by a group of students, and since no study is perfect, we have addressed your comments by modifying our limitation section.

4. you introduce the relevance of primary dysmenorrhea, but you don't mention it anymore in the paper in the results or discussion sections. How did you evaluate the apps for primary dysmenorrhea only? How did you know if the app was used for secondary dysmenorrhea either? Please explain the role of primary dysmenorrhea in the evaluation of the apps in the results of method section.

Response 4: We thank you for this valuable comment. Our research focuses basically on the evaluation of the Mobile apps and it was not aimed at distinguishing between primary and secondary dysmenorrhea. As mentioned in the previous comment, our study was explicitly designed for evaluation of MTAs aimed at managing primary dysmenorrhea. This was established by selecting only those apps that, based on their descriptions, were intended to help manage menstrual symptoms—specifically those relevant to PD. The focus on PD comes from its definition: painful menstrual cramps in women with no identifiable pelvic pathology, a condition that predominantly affects young women. Moreover, management of secondary dysmenorrhea requires diagnosis followed by physicians’ consultation. Hence, it is understood by the description of the apps and the understanding of the diseases that the selected app users are affected by PD. In addition, since our approach was only to evaluate those MTAs which are designed to manage PD, we didn’t mention further discussion of PD elsewhere by default.

5. Finally, what is your conclusion? This links back to your question. What is the answer to your question, based on the results? I don’t understand what I should have learned from your paper and how that can influence future research. It seems now that your conclusion is mainly for app developers, but your readers are mainly researchers, not developers.

Response 5: The conclusion has been revised and clarified. In a nutshell, this study answers its research question by emphasizing that while MTAs are a promising tool for menstrual health management, current versions fall short in delivering evidence-based exercise content for PD. This gap opens several avenues for future research aimed at enhancing clinical effectiveness, user engagement, and cultural relevance, thereby ultimately improving women's health outcomes. We are sure that our data will help the researchers in coming across potential challenges, and these information with these researchers carve the pathways for developers of MTAs.

Reviewer #2: Overall, the manuscript presents a well-executed study with significant contributions to the field of mHealth and women’s health. The methodology is rigorous, and the findings are well supported by data. However, certain areas require further refinement. The research question should be more explicitly stated in the introduction, and justification for methodological choices, such as the exclusion of lower-rated apps, should be provided. Greater emphasis on inter-rater reliability would enhance the validity of the app evaluations. The discussion would benefit from more direct comparisons with prior studies, and the results could be presented more effectively with visual aids. The conclusion should incorporate broader policy implications and clearly define future research directions.

Response: Dear Sir, thank you so much for taking time to review and provide feedback that is insightful. Your comments definitely helped us improve our manuscript. We made following changes:

1. The research question has been stated at the end of the introduction.

2. The exclusion criteria have been explained in methods.

3. Since our study is first of its kind in comparing the MTAs for PD, we are short of comparative studies to be included in the discussion section.

4. Graphical representation of mean app quality score has been done.

---

## [Decision Letter · Decision Letter 1]

13 Apr 2025

PONE-D-24-49548R1Mobile applications available in Saudi Arabia for the management of Primary Dysmenorrhea: a quality review and content analysisPLOS ONE

Dear Dr. alwhaibi,

Thank you for submitting your manuscript to PLOS ONE. After careful consideration, we feel that it has merit but does not fully meet PLOS ONE’s publication criteria as it currently stands. Therefore, we invite you to submit a revised version of the manuscript that addresses the points raised during the review process.

We look forward to receiving your revised manuscript.

Kind regards,

Mukhtiar Baig, Ph.D.

Academic Editor

PLOS ONE

Reviewers' comments:

Reviewer's Responses to Questions

**Comments to the Author**

1. If the authors have adequately addressed your comments raised in a previous round of review and you feel that this manuscript is now acceptable for publication, you may indicate that here to bypass the “Comments to the Author” section, enter your conflict of interest statement in the “Confidential to Editor” section, and submit your "Accept" recommendation.

Reviewer #1: (No Response)

2. Is the manuscript technically sound, and do the data support the conclusions?

Reviewer #1: Yes

3. Has the statistical analysis been performed appropriately and rigorously? 

Reviewer #1: Yes

4. Have the authors made all data underlying the findings in their manuscript fully available?

Reviewer #1: Yes

5. Is the manuscript presented in an intelligible fashion and written in standard English?

Reviewer #1: Yes

6. Review Comments to the Author

Reviewer #1: Thank you for replying to my previous questions. You provide quite complete answers to my questions, yet make hardly any significant changes in the manuscript. The explanations you give me should be given to the reader and, therefore, be part of the paper, not only the reply.

Please provide references and evidence for your explanations and add them to the paper. For example, if I ask you why you focus on physical exercise, and you answer: "physical therapy approaches, particularly exercise-based interventions, have demonstrated significant effectiveness in reducing primary dysmenorrhea pain. " but with no evidence to back your statement up. You need to build your arguments to your research question: "The evaluation of the quality Mobile apps for PD management in KSA in of MTAs available to users in Saudi Arabia on both iOS and Android platforms and examined the

efficacy of the exercise regimens provided as interventions for managing PD symptoms". I still don't understand your question. Are you looking at the quality of apps or the effectiveness of exercise? I also do not think your research question is in proper English.

Another example, I ask you why focus on PD? The answer is: "Our research limits the evaluation to primary dysmenorrhea because it is defined by the absence of pelvic abnormalities, predominantly affects young women, and allows for a more focused and consistent methodological approach". This does not explain why, only what PD is. Can you give me solid arguments on why you focus only on PD, based on references from current literature and please add it to your paper as well.

The same goes for all questions, except nr 5, where you actually did make some minor changes to the conclusion.

7. PLOS authors have the option to publish the peer review history of their article (what does this mean? ). If published, this will include your full peer review and any attached files.

**Do you want your identity to be public for this peer review?** For information about this choice, including consent withdrawal, please see our Privacy Policy .

Reviewer #1: **Yes: ** RA de Leeuw

---

## [Author Response · Author response to Decision Letter 2]

1 May 2025

Response to Reviewer’s Comments

Manuscript Title: Mobile applications available in Saudi Arabia for the management of Primary Dysmenorrhea: a quality review and content analysis

Manuscript ID: PONE-D-24-49548R1

Journal: PLOS ONE

Dear Editor,

We sincerely thank you and the reviewer for the valuable feedback provided on our manuscript. We have carefully considered all comments and substantially revised the manuscript to address each point raised. Below is a detailed, point-by-point response to Reviewer #1’s comments. All additions and modifications have been incorporated into the revised manuscript and are highlighted using track changes.

Reviewer’s Comments Our Responses Page #

1 You provide complete answers to my questions, yet make hardly any significant changes in the manuscript. The explanations you give me should be given to the reader and, therefore, be part of the paper We thank the reviewer for this important reminder. We have now integrated all previously provided explanations directly into the manuscript in relevant sections (Introduction, Rationale, Methods, and Discussion). These additions include references and clearer argumentation regarding the study’s rationale, scope, and contribution. Every major reviewer point is now explicitly and transparently addressed in the manuscript body. -

2 I still don't understand your question. Are you looking at the quality of apps or the effectiveness of exercise?

We have reworded both the abstract and the final paragraph of the introduction to clarify the dual aim of the study. Specifically, we now state that our study evaluates:

(1) the overall quality of MTAs using the Mobile Application Rating Scale (MARS), and

(2) the alignment of exercise-related content with established physical therapy guidelines, particularly the FITT (Frequency, Intensity, Time, and Type) principle.

We do not claim to assess the clinical efficacy of the apps or their exercise interventions. Page 7

3 Why focus on PD? The answer you give is descriptive, not analytical.

We have added a new paragraph in the “Dysmenorrhea and its Management” section to justify our focus on primary dysmenorrhea (PD). Our rationale includes its high prevalence (up to 89%), its suitability for non-invasive self-management, and the methodological advantage of studying a condition that does not require medical diagnosis. These points are now referenced and contextualized within the research literature. Page 4

4 Provide references and evidence for your explanations… You say exercise helps with PD but give no evidence. We have added multiple references to support the use of exercise-based interventions for primary dysmenorrhea. These citations are now included in the “Physical Therapy Approach for PD” section, which highlights the evidence-based rationale for evaluating exercise content in mHealth apps for PD management. Page 4

5 Why focus on exercise? Please give solid arguments supported by literature.

A new paragraph has been added at the end of the “Physical Therapy Approach for PD Management” section. This explains that exercise-based strategies are evidence-backed, safe, and cost-effective, and are already included in many non-pharmacological treatment protocols for PD. Supporting references have been added. Page 4

6 PD isn’t mentioned later. How did you ensure these apps were only for PD and not secondary dysmenorrhea?

We have now included a methodological clarification in the “Eligibility Criteria” section explaining that we excluded apps referring to endometriosis, fibroids, or other pathological pelvic conditions. Our inclusion criteria relied on keyword searches and app descriptions that specified menstrual pain or cramps without underlying medical conditions. Page 9

7 Why Arabic? You did not exclude anything based on the Arabic nature of your study.

We have clarified this point in the Methods and Limitations sections. While the intent was to include Arabic-language apps, our search strategy did not use Arabic-only language as an exclusion criterion. The majority of high-rated apps in Saudi Arabia are in English, and we included them to reflect actual user availability and experience. We now clearly state that this is a limitation in serving monolingual Arabic speakers and highlight it as a gap in the market. Page 10, page 23

8 What is your conclusion? I don’t understand what I should have learned.

We have rewritten both the opening paragraph of the Discussion section and the Conclusion. These now clearly summarize the dual findings: that most MTAs offer acceptable app quality but fail to meet clinical standards for exercise content. We emphasize the need for interdisciplinary development, localization, and evidence-based integration in MTA design for managing PD. The implications for app developers, physical therapists, and women’s health stakeholders are now clearly articulated. Page 23

We hope that the revisions made in response to your insightful feedback have strengthened our manuscript and clarified our contributions to this important area of mHealth research. We are grateful for the reviewer’s careful and constructive critique, which has significantly enhanced the clarity, scientific rigor, and relevance of our work.

We respectfully submit the revised manuscript for your further consideration.

Sincerely,

Dr. Tahani J. Alahmadi

Princess Nourah bint Abdulrahman University

Email: Tjalahmadi@pnu.edu.sa

---

## [Editor Report · Decision Letter 2]

18 May 2025

Mobile applications available in Saudi Arabia for the management of Primary Dysmenorrhea: a quality review and content analysis

PONE-D-24-49548R2

Dear Dr. Al whaibi,

We’re pleased to inform you that your manuscript has been judged scientifically suitable for publication and will be formally accepted for publication once it meets all outstanding technical requirements.

Kind regards,

Mukhtiar Baig, Ph.D.

Academic Editor

PLOS ONE

---

## [Editor Report · Acceptance letter]

PONE-D-24-49548R2

PLOS ONE

Dear Dr. alwhaibi,

I'm pleased to inform you that your manuscript has been deemed suitable for publication in PLOS ONE. Congratulations! Your manuscript is now being handed over to our production team.

Kind regards,

on behalf of

Professor Mukhtiar Baig

Academic Editor

PLOS ONE